# Patient Management in the Emergency Department during a COVID-19 Pandemic

**DOI:** 10.3390/healthcare10081456

**Published:** 2022-08-03

**Authors:** Marlena Robakowska, Anna Tyrańska-Fobke, Katarzyna Pogorzelczyk, Joanna Synoweć, Daniel Ślęzak, Piotr Robakowski, Patryk Rzońca, Paweł Prędkiewicz

**Affiliations:** 1Department of Public Health and Social Medicine, Medical University of Gdańsk, 80-210 Gdansk, Poland; mrobakowska@gumed.edu.pl; 2Independent Researcher, 8200 Aarhus, Denmark; kpogorzelczyk@gmail.com; 3Independent Researcher, 80-210 Gdansk, Poland; joanna@synowec.pl; 4Department of Medical Rescue, Medical University of Gdańsk, 80-210 Gdansk, Poland; daniel.slezak@gumed.edu.pl; 5Department of Safety Sciences, University of Gdansk, 80-309 Gdansk, Poland; piotr.robakowski@ug.edu.pl; 6Department of Human Anatomy, Faculty of Health Sciences, Medical University of Warsaw, 02-004 Warsaw, Poland; przonca@wum.edu.pl; 7Department of Finance, Faculty of Economics and Finance, Wrocław University of Economics, 53-345 Wrocław, Poland; pawel.predkiewicz@ue.wroc.pl

**Keywords:** management, emergency department, COVID-19, pandemic

## Abstract

In the wake of the COVID-19 pandemic, international action has been taken to prevent the spread of the disease. The aim of this study is to establish the impact of the COVID-19 pandemic on emergency department utilization patterns in Poland. It was established that age (among COVID-19 positive patients) has a large influence on the occurrence of a mental illness or disorder. It has been proven that the older the person (patients diagnosed with U07.1), the more often mental diseases/disorders are diagnosed (*p* = 0.009–0.044). Gender decides the course of hospitalization to the disadvantage of men (*p* = 0.022). Men diagnosed with U07.1 stay much longer in specialized long-term care units. Lower-aged patients have a shorter hospitalization time (up to the age of 29; *p* = 0.017). The COVID-19 pandemic has placed healthcare systems, their staff, and their patients in an unprecedented situation. Our study showed changes in the number and characteristics of patients visiting the ED during COVID-19. Despite the shift in the center of gravity of health system functioning to the treatment of SARS-CoV-2 infected patients, care must be taken to ensure that uninfected patients have access to treatment for cardiovascular, mental health, oncological, and other diseases.

## 1. Introduction

An infectious disease caused by the coronavirus SARS-CoV-2 began as an epidemic in Wuhan, China, in November 2019. It was declared a pandemic by the WHO on 11 March 2020. By 8 March 2022, more than 448 million cases of the disease had been reported in 188 countries and territories, with over 6 million patients dead [1,2]. In the wake of the pandemic, international action has been taken to prevent the spread of the disease. Quarantines and curfews were introduced, travel was restricted, and many cultural, religious, and sporting events were canceled (e.g., the 2020 Olympic Games in Tokyo). Some countries introduced restrictions on border traffic or completely closed their borders [3]. Restrictions were introduced on international flights and border crossings, and checks at airports and railway stations were implemented. Schools and universities in most countries switched to distance learning, and, where possible, employees were moved from offices to their homes [3,4]. The pandemic caused global economic and social disruption, triggering the biggest recession since the Great Depression.

In Poland, the COVID-19 pandemic exposed the collapse of health policy. In its initial phase, there was a lack of testing and basic protective measures. Local people held collections to buy masks and protective gowns, and beauty parlors and tattooists donated their disposable gloves to hospitals. Chaos, uncoordinated actions, and procedural problems were evident ‘missed opportunities’ during the initial wave of the pandemic [5]. Government action was in response to emergencies rather than a planned action. Issues such as the underfunding of the system, significant shortages in the health professions, and the need for reform became glaring and clearly visible. All efforts were concentrated on the fight against the virus, forgetting about ‘standard’ patients. As a result, waiting times for scheduled admissions and specialized medical services increased significantly. The initial phase of the fight against the pandemic was similar in the United Kingdom, for example. The British, after the first phase of “downplaying” COVID-19, tried in a rather chaotic way to bring order to society and stop the disease from spreading. It can be assumed that the vast majority of systems reacted in a similar way—initially with some incredulity and then rapidly and unsystematically introducing norms to halt the development of the pandemic. Poland is no exception in this respect, but due to earlier systemic problems, its situation in the fight against the disease was on the losing end from the beginning [5].

The aim of this study is to show the impact of the COVID-19 pandemic on emergency department utilization patterns in Poland by a correlation between metrics, information on hospitalization, and COVID-19, including comorbidities that may affect the management of an emergency patient (in the emergency department) during a pandemic.

## 2. Materials and Methods

The study used existing data on patients admitted to the clinical emergency department of the University Clinical Center in Gdańsk—the largest and most specialized hospital in the Pomeranian Voivodeship in Poland. Data of patients admitted to clinical ED in the period from January to June 2021 with a confirmed positive PCR test result for SARS-CoV-2 infection were analyzed.

The most important criterion for including data in the study was a positive PCR test result for the presence of the SARS-CoV-2 virus in patients admitted to the clinical emergency department of the analyzed facility during the study period. Data from other patients admitted in the same period without PCR confirmed SARS-CoV-2 infection were not included in the study.

Cross-sectional data, which formed the basis of the study, were subjected to basic data analysis and statistical analysis in terms of relationships, including gender, age, and diagnoses. Chi2 distribution tests were performed. Microsoft Excel 2019 (Redmond, Washington, DC, USA) and Stata 15.1 (StataCorp, College Station, TX, USA) were used to implement them [6,7]. The threshold of statistical significance was set at 0.05 (*p* = 0.05).

## 3. Results

### 3.1. Analyzed Groups

On the basis of the information included in the analysis, it can be shown that the group of respondents consisted of 1252 patients, where 44.8% were women (*n* = 561) and 55.2% were men (*n* = 691), respectively. The characteristics of the analyzed population are presented in Table 1.

The mean age of the patients was 48.92 for those with COVID-19 infection as a comorbid diagnosis and 53.17 for patients with SARS-CoV-2 as the primary diagnosis, which is shown in Table 2.

### 3.2. Diagnoses

The preliminary analysis showed that the most common diagnosis assigned to the patients was N.18, i.e., chronic renal failure, with a total of 1252 patients (Table 3). The next diseases on the list were U07.1 and Z51.1, i.e., pneumonia due to SARS-associated coronavirus and encounter for antineoplastic chemotherapy and immunotherapy, respectively. The list also includes a disease exclusively attributed to women—C56, which is ovarian cancer (*n* = 172).

The table presented above illustrating the most common diseases in the general group of respondents does not translate into specific age groups. The most frequently recorded diagnoses for the oldest groups, i.e., 90+, 80–89, and 70–79 years of age, were I20.0, Z74.3, and I82.9, respectively, i.e., unstable angina, need for continuous supervision, and embolism and thrombosis of unspecified vein. It is worth noting that for the last of the above-discussed groups, 70–79 years of age, group I diseases were the most common, in total, about 26% of cases. Internal hemorrhoids without complication, I84.2, came in second. A detailed list is presented in Table 4. What can be mentioned, however, is the characteristics of the 30–39 age group, in which the information was worth considering broken down into men and women. Among the female sex, the most common diagnosis was O.36, which according to the ICD10 classification, is maternal care for other fetal problems, which is probably related to the age of the woman, which is more and more often referred to as late in terms of getting pregnant. The second most common disease in this group, both for men and women, was D64.8, other specified anemias. The age group of 1–9, the youngest patients, most often struggled with the problem of endocrine disorder, unspecified, thrombocytopenia, unspecified, and precocious puberty. The first 2 of the aforementioned diseases mainly affected children aged 1–4 years.

### 3.3. Hospitalizations

The COVID-19 pandemic has contributed to the need to introduce changes to the collection or presentation of data related to hospitalization. From the beginning of the COVID-19 period, it has been one of the most common diagnoses or causes of hospitalization. From the perspective of the following study, it was surprising to find out about the highest levels of operation of specific hospital departments. Which of them had the highest mean hospitalization rates during the worst pandemic crisis? The analysis showed that the adult psychiatry department was number one on the list. The average length of stay in the ward was nearly 51 days. The second on the list was the unit dedicated to COVID-19. The average stay here lasted 14.19 days for basic hospitalization and 9.42 days for coexisting COVID-19 cases. A detailed list is presented in Table 5.

Another compilation prepared for the purposes of this study is information on the average length of hospitalization in relation to specific diseases according to the ICD10 classification. Among women, the longest hospital stay was expected in bipolar disorder, current episode mixed, which is reflected in the results discussed above (the adult psychiatry unit had the longest hospitalizations). In the group of men, patients spent the most time in the case of developing other nonrheumatic aortic valve disorders (*n* = 39.95 days). The disease with number U07.1, COVID-19, virus identified, was associated with the need for a long stay in the hospital (compared to other diseases; Table 6) for women and men, 4.26 days and 12.95 days, respectively.

As mentioned above, apart from the use of descriptive statistical methods to analyze the data included in the study, a statistical analysis was also performed in terms of correlation. The main focus was on finding the relationship between metric data, information on hospitalization, and COVID-19. It was established that age (among COVID-positive patients) had a large influence on the occurrence of a mental illness or disorder. It was proven that the older the person (patients diagnosed with U07.1), the more often mental diseases/disorders are diagnosed (*p* = 0.009–0.044). Interestingly, and also based on the earlier discussion of the results (the most common diseases in specific age groups), COVID+ women in the 30–36 age group more often struggled with the problems of maintaining pregnancy. The result of the analysis was at the level of *p* = 0.036. The correlation analysis also showed some dependencies regarding the hospitalization itself—the patient’s characteristics and the length of stay in the hospital. Namely, gender determined the course of hospitalization to the disadvantage of men (*p* = 0.022). Men diagnosed with U07.1 stayed much longer in specialized long-term care units. Lower age determined the shorter hospitalization time (up to the age of 29; *p* = 0.017). Our study showed changes in the number and characteristics of patients visiting the ED during COVID-19.

## 4. Discussion

The COVID-19 pandemic outbreak has not resulted in the absence of other diseases [8]. On the contrary, it has become an additional threat to chronically ill and immunocompromised individuals [9,10]. The purpose of this study was to determine which diseases were most common among COVID-positive individuals.

The present study showed that the older a COVID+ patient was, the more often he or she was diagnosed with a mental health condition. The COVID-19 outbreak has caused fear and anxiety. Preventive measures, such as isolation and quarantine, can cause fear, anxiety, and uncertainty in patients, resulting in an increase in stress-related illnesses while exacerbating pre-existing mental disorders [10,11]. Individuals with mental disorders are more likely to have emotional reactions related to a COVID-19 outbreak, which can lead to relapse and worsening of the condition. Individuals with psychiatric disorders are more susceptible to stress compared to the general population [12]. Recurrence of serious mental disorders can result in poor hygiene, inability to implement social distancing or other preventive strategies, failure to report in a timely manner or obtain medical attention, and failure to adhere to expected treatment [13,14]. Sukut and Ayhan Balik emphasize that as a result of the traffic restrictions and social isolation applied in the wake of the COVID-19 pandemic, outpatients with serious mental illness have difficulty maintaining treatment. Therefore, prevention measures taken in the wake of the outbreak may cause relapse and behaviors such as hyperactivity, agitation, self-harm, and others. Therefore, these prevention measures, which also increase the risk of suicide and negative emotions, may lead to decreased well-being for individuals with serious mental illness. As a result, these individuals may experience feelings such as loneliness, denial, anxiety, depression, insomnia, and a sense of hopelessness that may reduce the effectiveness of treatment. Social stressors that can trigger serious mental disorders such as depression or anxiety in previously healthy people, such as health anxiety, fear of death, loss of loved ones, loss of social ties, loss of jobs, and homelessness, can cause more serious problems in people with serious mental illness [15].

Results showed that COVID + women in the 30–36 age group more often struggled with the problems of maintaining pregnancy. A systematic review by Zgliczyńska and Kosińska–Kaczyńska presents a study on 108 pregnant women with SARS-CoV-2 infection [14]. The most common clinical symptoms in this group were fever (68%) and cough (34%), followed by fatigue (13%), dyspnea (12%), and diarrhea (6%). The condition of 3 out of 108 (3%) analyzed patients required their admission to the ICU. There was one case of neonatal death and one case of intrauterine fetal death, but no deaths among pregnant women [16]. Also, Breslin et al. published the results of a study involving one of the largest groups of pregnant patients with SARS-CoV-2 infection; 43 women with confirmed real-time polymerase chain reaction (RT-PCR), who were reported to 2 hospitals in New York. In the study group, only 7% of women were admitted to the hospital initially due to COVID-19 symptoms, while almost half of them were due to obstetric reasons. The remaining 51% of patients, due to their well-being, did not initially require hospitalization and received an outpatient consultation. A total of 14 out of 43 patients had no symptoms on admission, but the condition of 2 of them deteriorated rapidly in the following days, which necessitated hospitalization in the ICU due to respiratory failure [17]. The most common symptoms were dry cough (66%), fever (48%), and muscle aches (39%). Headache (28%), dyspnoea (24%), and chest pain (17%) were less frequent. Using the criteria for assessing the severity of the disease proposed by Wu et al., in the entire study group, 86% of patients underwent the infection mildly, 9% severely, and 5% critically [18]. Based on the above information, it can be concluded that COVID-19 was not an immediate threat to pregnancy in most cases, but affected pregnant women required detailed observation.

This paper shows that gender and age matter in the length of hospitalization of COVID+ patients. Women and young people under 29 years of age have a longer hospital stay than men and people over 30 years of age.

It is worth emphasizing that early use of hydroxychloroquine in mild patients is hypothesized to be beneficial in reducing patients’ hospitalization [19]. After validated clinical trials, early administration of hydroxychloroquine, with antibiotics or zinc, where appropriate, could become the first line of a global strategic response to this and subsequent COVID-19 epidemics, allowing for a significant reduction in preventive measures and hospitalization needs [19].

In a report published by the Chinese Centre for Disease Control and Prevention [10], 87% of patients were aged 30–79 years, with only 3% of patients aged 80 years and older. Of note is the significant increase in mortality in elderly patients. In the cited Chinese report, the overall mortality rate for COVID-19 was 2.3%, but in the group of patients aged 70 to 79 years, it was 8%, while in the group ≥80 years, it reached 14.8%. Italian researchers, who also published their COVID-19 mortality data and compared them with the WHO mortality report in China, present the hypothesis that the older the population, the higher the overall mortality rate. The overall mortality rate in Italy was 7.2%, where patients aged over 70 years account for up to 37.6% of all cases, compared with 11.9% in China. In addition, a sizeable group of patients in Italy are in their 90s (such a group is not specified in Chinese reports). This age group also has the highest mortality rate (22%). The older age of patients is cited as an independent risk factor for both death and severity of infection [18,20]. Additional factors affecting the severity of the course of SARS-CoV-2 infection are chronic diseases, the prevalence of which increases with age. Although they do not seem to affect the risk of infection with the new coronavirus, diseases such as hypertension, ischaemic heart disease, or diabetes are two (hypertension and diabetes) or even three times (ischaemic disease) more frequent in patients hospitalized in the intensive care unit [21]. Moreover, the risk of death has been demonstrated to be significantly higher in patients with the above-mentioned diseases as well as with chronic obstructive pulmonary disease or chronic kidney disease [22].

The study results presented above note the astonishingly low number of admissions and hospitalizations for cardiovascular disease (CVD), which are closely related to the reigning COVID-19. SARS-CoV-2 significantly affects the cardiovascular system. Microvascular damage, endothelial dysfunction, and thrombosis resulting from viral infection or indirectly related to an enhanced systemic inflammatory and immune response are hallmarks of severe COVID-19. Pre-existing cardiovascular disease and viral infection are associated with myocardial damage and worse outcomes. The vascular response to cytokine production and the interaction between coronavirus type 2 acute respiratory syndrome (SARS-CoV-2) and angiotensin-converting enzyme receptor 2 may lead to a significant reduction in cardiac contractility and subsequent myocardial dysfunction [23]. In the present study, only among patients aged 50–59 years was this the most common diagnosis. This should not be interpreted as a lack of affected individuals, but rather that CVDs ‘hid’ under the umbrella of COVID-19. The study by Shi et al. [24] shows that abnormal elevation of cardiac injury biomarkers is widely present in patients with COVID-19 and is likely to be associated with myocarditis associated with infection, right heart overload, and/or ischemia [24]. The analysis shows that biomarkers of cardiac injury are closely associated with disease progression and prognosis. In a cohort study of COVID-19 patients, cardiac injury occurred in 19.7% of patients during hospitalization and was an independent risk factor for in-hospital mortality [24]. During the SARS-CoV-2 pandemic, most patients with CVD symptoms were COVID+; hence, their classification was under this diagnosis. This trend is likely to be reversed once the primary morbidity has subsided and long-term cardiovascular outcomes of COVID-19 have developed. Xie, Bowe, and Al-Aly [25] used national healthcare databases from the US Department of Veterans Affairs to build a cohort of 153,760 individuals with COVID-19, as well as 2 sets of control cohorts with 5,637,647 (contemporary controls) and 5,859,411 (historical controls) individuals, to estimate risks and 1-year burdens of a set of pre-specified incident cardiovascular outcomes. Their study showed that CVD occurs both immediately after COVID infection (up to 30 days) and long-term afterward [25]. Immediate symptoms are usually treated during the first hospitalization and are classified as U07.1 COVID-19, virus identified. The study also showed that the risk and annual burden of cardiovascular disease in acute COVID-19 survivors are significant. The results showed that the adjusted incidence rates of cardiovascular events after exposure to COVID-19 were significantly higher than those in the pre-exposure period (incidence rates for all cardiovascular events were significantly higher than 1 and showed a graded increase depending on the severity of the acute phase of the disease [25].

Another reason for the low number of CVD patients may be the public’s fear of visiting the emergency department during the pandemic. During the peak incidence period, patients were discouraged from visiting the emergency room because of the heavy burden on hospitals by people with serious COVID-19 infection or the fear of being infected by other patients or hospital staff. Additionally, people who have already undergone COVID-19 in a hospital setting are reluctant to return to the ward with new symptoms, such as angina. A study by Sürme et al. [26] on 639 people shows that patients admitted to the emergency department were afraid of COVID-19. Fear of COVID-19 was significantly higher in women aged 57 years and older, with lung disease, COVID-19 symptoms, and children. Lung disease, female gender, and fear of COVID-19 were statistically significantly correlated with the occurrence of fear. This demonstrates the real public health problem of the inability to diagnose and treat diseases early, which is particularly important in cardiovascular diseases. In terms of managing a medical facility in the time of a pandemic, the authors indicate four key areas that require immediate actions to improve the functioning of the health and behavioral system public health of the population. Based on the analysis of nine crisis levers of the business model of A. Osterwalder, Y. Pigneur, four main areas requiring immediate intervention were identified. These are the development of medical personnel, development of computerization and digital competencies, optimization of revenues and costs, indication of decision-making centers—appropriate division of tasks and responsibilities. In these areas, the Polish health care system has failed; they require urgent strengthening and long-term strategic solutions, which will translate into the optimization of activities aimed at securing the health of the society. It turns out that during the pandemic, it was not costs that turned out to be the biggest problem, but above all, staff shortages, lack of access to health services, or equipment and organizational shortages. The time of the pandemic showed that researchers’ theories that quality and not cost is the most important were proved correct. According to M.E. Porter and E. Teisberg, despite the limitations, the healthcare system can achieve excellent results both in the area of quality and effectiveness, which, however, often requires transformation and implementation of new competition rules aimed at increasing the value created [27].

The study has several limitations. The most important ones include the analysis of existing hospital data, which in some respects could be incomplete. Another limitation is the analysis of patients from only one emergency department. However, the fact that it is the emergency department of the largest hospital in the region with the highest reference value speaks in favor of the study.

## 5. Conclusions

The COVID-19 pandemic has placed healthcare systems, their staff, and their patients in an unprecedented situation. There have been changes in the number and characteristics of patients visiting EDs during COVID-19. In this era of ever new infections, special care must be given to the elderly, patients with chronic diseases, people with mental disorders, and pregnant women, as they are the most at risk from the coronavirus. Myocardial damage and inflammation are often observed in infected individuals, which is associated with increased mortality. For this reason, in patients with COVID-19 or with suspected infection, it is important to look not only for respiratory symptoms but also for cardiovascular symptoms. Despite the shift in the center of gravity of health system functioning to the treatment of SARS-CoV-2 infected patients, care must be taken to ensure that uninfected patients have access to treatment for cardiovascular, mental health, oncological, and other diseases.

## Figures and Tables

**Table 1 healthcare-10-01456-t001:** Characteristics of the analyzed population.

Age Group	Number of Patients and %
90+	47 (3.75%)
80–89	113 (9.03%)
70–79	208 (16.61%)
60–69	217 (17.33%)
50–59	151 (12.06%)
40–49	117 (9.35%)
30–39	156 (12.465)
20–29	129 (10.3%)
10–19	46 (3.67%)
1–9	68 (5.43%)
**Sex**	
Women	561 (44.8%)
Men	691 (55.2%)

**Table 2 healthcare-10-01456-t002:** Average age of analyzed group.

Average of Patient Age	Age	
	Coexisting	Essential
Women	50.99	52.82
Men	47.12	53.51
Total	48.92	53.17

**Table 3 healthcare-10-01456-t003:** The most popular diagnoses in the group of patients by gender.

ICD-10 Code	Women	Men	Total
N18 Chronic renal failure	625	627	1252
U07.1 COVID-19, virus identified	350	515	865
Z51.1 Chemotherapy session for neoplasm	322	375	697
Z03.9 Observation for suspected disease or condition, unspecified	178	120	298
C91.0 Acute lymphoblastic leukemia	43	169	212
C56 Malignant neoplasm of ovary	172	0	172

**Table 4 healthcare-10-01456-t004:** Recognitions among age groups—the most common.

Age Group (Women and Men)	ICD-10	
90+	I20.0 Unstable angina N30.0 Acute cystitis S72.1 Pertrochanteric fracture	
80–89	Z74.3 Need for continuous supervision I74.4 Embolism and thrombosis of arteries of extremities, unspecified I63.8 Other cerebral infarction	
70–79	I82.9 Embolism and thrombosis of unspecified vein I84.2 Internal rectal bleeding tumors without complications K57.9 Diverticular disease of the intestine, part unspecified, without perforation or abscess	
60–69	C49.4 Connective and soft tissue of the abdomen I10 Essential (primary) hypertension C48.0 Retroperitoneum	
50–59	C90 Multiple myeloma and malignant plasma cell neoplasms I11 Hypertensive heart disease C56 Malignant neoplasm of ovary	
40–49	R49 Voice disturbances R07 Pain in throat and chest N08.5 Glomerular disorders in systemic connective tissue disorders	
30–39	WOMEN 036 Maternal care for other known or suspected fetal problems D64.8 Other specified anemias	MEN D64.8 Other specified anemias
20–29	D64.8 Other specified anemias E10.9 Type 1 diabetes mellitus without complications O80.0 Spontaneous vertex delivery	
10–19	C81.1 Nodular sclerosis (classical) Hodgkin lymphoma Z94.8 Other transplanted organ and tissue status R76.8 Other specified abnormal immunological findings in serum	
1–9	E34.9 Endocrine disorder, unspecified D69.6 Thrombocytopenia, unspecified E30.1 Precocious puberty	

**Table 5 healthcare-10-01456-t005:** Average time of hospitalization (in days) of specific hospital wards broken down into coexisting and essential diagnoses in relation to the diagnosis associated with COVID-19.

Hospital Ward	Coexisting Diagnosis Average Time of Hospitalization (in Days)	Essential Diagnosis Average Time of Hospitalization (in Days)
Department of Adult Psychiatry		50.96
Department of Anaesthesiology and Intensive Therapy		0.65
COVID-19 Clinic	9.42	14.19
Department of Cardiac Surgery and Vascular Surgery		3.06
Department of Endocrinology and Internal Medicine	11.93	
II Department of Cardiology		4.90
Department of Adult Neurology		12.66
First Department of Cardiology		8.66
Department of Nephrology, Transplantology, and Internal Diseases	11.06	7.5
Clinical Emergency Department	0.52	4.09
Department of Oncology and Radiotherapy		7.32
Department of Hypertension and Diabetology		3.6
Department of Allergology and Pneumology		6.85
Department of Hematology and Transplantology		0.2
Department of Gynecology, Oncological Gynecology, and Gynecological Endocrinology		3.77
Department of Urology		0.6
Department of Pediatrics, Hematology, and Oncology	0.1	0.17
Department of Pediatric Cardiology and Congenital Heart Defects		1.17
Department of Otolaryngology with the Department of Maxillofacial Surgery		0.94
Department of Pediatrics, Diabetology, and Endocrinology		0.21
Department of Ophthalmology		0.33
Average	7.35	5.71

**Table 6 healthcare-10-01456-t006:** Average time of hospitalization in days.

ICD-10 Code	Women	Men	Total
F31.6 Bipolar affective disorder, current episode mixed	82.98		82.98
I35.8 Other aortic valve disorders		39.95	39.95
F10.2 Mental and behavioral disorders due to use of alcohol		18.94	18.94
G61.0 Guillain-Barré syndrome		18.9	18.9
I50.0 Congestive heart failure		15.08	15.08
I63.3 Cerebral infarction due to thrombosis of cerebral arteries	13.25		13.25
N18.9 Chronic kidney disease, unspecified	12.47		12.47
C24.9 Biliary tract, unspecified		11.93	11.93
D64.9 Anemia, unspecified	9.65	11.93	10.79
O99.2 Endocrine, nutritional, and metabolic diseases complicating pregnancy, childbirth, and the puerperium	10.1		10.1
N18.0 Chronic renal failure	14.7	0.3	9.9
D64.8 Other specified anemias	9.65		9.65
O99.5 Diseases of the respiratory system complicating pregnancy, childbirth, and the puerperium	9.47		9.47
E66.0 Obesity due to excess calories		9.42	9.42
U07.1 COVID-19, virus identified	4.27	12.96	8.85
O36.7 Maternal care for viable fetus in abdominal pregnancy	8.83		8.83
I63.9 Cerebral infarction, unspecified	13.25	5.83	8.30
Z51.0 Radiotherapy session		7.32	7.32
M35.9 Systemic involvement of connective tissue, unspecified	6.97		6.97
I50.9 Heart failure, unspecified		4.52	4.52
I47.1 Supraventricular tachycardia	3.91		3.91
O82.1 Delivery by emergency cesarean section	3.81		3.81
I12.0 Hypertensive renal disease with renal failure		3.69	3.69
J15.8 Other bacterial pneumonia		3.51	3.51
I50.1 Left ventricular failure		3.51	3.51
Z03 Medical observation and evaluation for suspected diseases and conditions, ruled out		3.06	3.06
I70.8 Atherosclerosis of other arteries		3.06	3.06
O80.0 Spontaneous vertex delivery	2.41		2.41

## Data Availability

Not applicable.

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
