# Peer review of "Patient Management in the Emergency Department during a COVID-19 Pandemic"

_healthcare, 2022, doi:10.3390/healthcare10081456_

Round 1

Reviewer 1 Report

1: It is recommended that the article be carefully read and check the grammar and spelling of the words. for example in

line 40: cancelled= canceled

line 52: planned=A planned

line 74: was = were  

line 82: respectively comes at the end of the sentence

Some sentences need a reference, such as line 41, which can use "10.18502/ijph.v50i4.5991".

Tables 1 and 2 are not referenced in the text of the article.

It is better to write the name of the disease in the tables next to the ICD-10 code.

The gender and border between the groups are not clear in Table 3, and the authors should design this table in a more understandable form.

Author Response

Thank you for all valuable comments. Please see the attachment.

Reviewer 2 Report

The study by Marlena Robakowska et al. reports a scenario of patient management in the emergency department based on retrospective data. I have the following comments, which may hopefully improve the quality.

·       The method section is suggested to improve: ethical committee approval is required for clinical studies involving patient data. Proper citation of software package used in the current study. What is the inclusion and exclusion criteria of the current study?

·       A summary table for all patients involved is suggested.

·       Regarding the hospitalization, early use of hydroxychloroquine at mild patients is hypothesized to be beneficial in reducing patients’ hospitalization (PMID: 33116907). Please elaborate on this in the discussion section.

·       In table 4, are patients in these departments are mutually exclusive to each other? It would be possible that patients can move from general ward into ICU ward.

·       Findings in the table 3 can be better shown with a figure.

Author Response

(The authors gave the same response as above.)

Round 2

Reviewer 2 Report

The authors have fully addressed my request, and I do not have any further comments.

Author Response

Thank you very much for your kind reply.